# A Review on Aging, Sarcopenia, Falls, and Resistance Training in Community-Dwelling Older Adults

**DOI:** 10.3390/ijerph19020874

**Published:** 2022-01-13

**Authors:** Filipe Rodrigues, Christophe Domingos, Diogo Monteiro, Pedro Morouço

**Affiliations:** 1ESECS—Polytechnic of Leiria, 2411-901 Leiria, Portugal; filipe.rodrigues@ipleiria.pt (F.R.); diogo.monteiro@ipleiria.pt (D.M.); 2Life Quality Research Center (CIEQV), 2040-413 Rio Maior, Portugal; christophedomingos@esdrm.ipsantarem.pt; 3Research Center in Sports, Health and Human Development (CIDESD), 5001-801 Vila Real, Portugal; 4Center for Innovative Care and Health Technology (ciTechCare), 2410-541 Leiria, Portugal

**Keywords:** aging, sarcopenia, fall, resistance training, elderly

## Abstract

As aging continues to grow in our society, sarcopenia and associated fall risk is considered a public health problem since falling is the third cause of chronic disability. Falls are negatively related to functionality and independence and positively associated with morbidity and mortality. The cost of treatment of secondary injuries related to falls is high. For example, one in ten fall incidents leads to bone fractures and several other comorbidities. As demonstrated by several experimental studies, adopting a more active lifestyle is critical for reducing the number of fall episodes and their consequences. Therefore, it is essential to debate the proven physical exercise methods to reduce falls and fall-related effects. Since muscle mass, muscle strength, bone density, and cartilage function may play significant roles in daily activities, resistance training may positively and significantly affect the elderly. This narrative review aimed to examine current evidence on existing resistance training using resistance machines and bodyweight or low-cost equipment for the elderly and how they are related to falls and fall-related consequences. We provide theoretical links between aging, sarcopenia, and falls linking to resistance training and offer practical suggestions to exercise professionals seeking to promote regular physical exercise to promote quality of life in this population. Exercise programs focusing on strength may significantly influence muscle mass and muscle strength, minimizing functional decline and risk of falling. Resistance training programs should be customized to each elderly according to age, sex, and other fundamental and individual aspects. This narrative review provides evidence to support recommendations for practical resistance training in the elderly related to intensity and volume. A properly designed resistance training program with adequate instructions and technique is safe for the elderly. It should include an individualized approach based on existing equipment (i.e., body weight, resistance machines). Existing literature shows that exercise performance towards 2–3 sets of 1–2 exercises per major muscle group, performing 5–8 repetitions or achieving intensities of 50–80% of 1RM, 2–3 times per week should be recommended, followed by training principles such as periodization and progression. Bearing this in mind, health and exercise professionals should combine efforts focusing on efficient strategies to reduce falls among the elderly and promote higher experiences of well-being at advanced stages in life.

## 1. Introduction

An aged population is no longer a future problem but of the present. Considering the numbers predicted by the United Nations indicating that the elderly population will reach 2 billion by 2050, global aging is bringing new realities for economic, social, and health systems in most countries [1].

The investment in the study of aging made perfect sense when society faced an aging population at the beginning of the 21st century. The growing worldwide aging is indisputable, in which the percentage of individuals over 65 years is the fastest-growing segment, and 125 million people are aged 80 years or older [2]. This aging stems from the increase in average life expectancy and the decrease in birth rate, causing an exponential rise in the elderly compared to the younger population [3]. The growth in the 65-year-old segment may also be related to advances in medical care and technology aiding in several daily activities, reducing daily physical demands. The world has witnessed enormous investments in treatment, whereas applying it to prevention may prove to be most cost-effective. For instance, disability was more predictive of mortality than multimorbidity for individuals aged 80 years and or older [4]. Survival curves demonstrate that more vital efforts should be made to make the quality of life a reality as it is more suitable to be preventing disability than treating non-communicable diseases [5]. The elderly want to live longer and gain or maintain quality of life during aging.

There are, however, several physical and mental implications of aging [6]. Several structural and functional changes occur during the aging process, such as loss of walking ability, muscle strength, balance, and flexibility [7]. Additionally, mental health is severely affected by neurological mechanisms related to aging [8], for example, vestibulocochlear degeneration [9] and reduced cognitive functions [10], as well as by social factors such as institutionalizations [9], dementia, and depression [10]. Research should focus on providing clear evidence on how healthy aging should be a priority, as recently expressed by the European Commission [3], shifting from active aging to healthy aging. Thus, a significant amount of research has been performed in recent years exploiting these domains’ interactions. Still, both physical activity (PA) and exercise have been presented in general terms, where there is a lack of understanding of the unique features that specific training types may provide.

As individuals age, physical fitness level decreases with advancing age at a uniform rate. The maintenance of flexibility generates force and power, and coordination is dependent on physical activity levels during previous stages in life, such as adulthood [6]. Hence, it is paramount to define effective interventions that promote physical activity, focusing on resistance training to reduce fall risk and fall-related injuries [11] since it is the most accessible training method for this community.

This narrative review aims to (i) conceptualize sarcopenia, both functional and neuromuscular; (ii) associate sarcopenia with falls and fall-related injuries; and (iii) discuss the effect of resistance training in the elderly population. The current study also aimed to review the resistance training characteristics used in previous studies to identify intervention effectiveness for improving or maintaining muscle mass and muscle strength. This review proposes several practical implications for exercise physiologists to consider during exercise prescription, based on intensity and volume.

### 1.1. Aging and Sarcopenia

This global panorama of worldwide aging requires knowledge that supports interventions that accompany associated consequences of aging are understood. Sarcopenia, characterized as losing muscle strength and mass in concert with biological aging, is one of the most important causes of functional decline and loss of independence in older adults. Characterized by progressive and generalized loss of skeletal muscle mass and strength, sarcopenia is strictly correlated with physical disability, poor quality of life, and death [12]. A 5–10% loss of muscle mass per decade after 50 years is estimated [13]. Given that muscle mass accounts for up to 60% of body mass, pathological changes to this physiological tissue can profoundly affect the older adult. The unbalanced process from anabolism and catabolism, favoring the latter, remains a challenge against falls, bone fractures, limited mobility, and increased mortality. Apoptosis begins if the muscle fiber size diminishes further to a necessary minimum. Furthermore, non-pharmacological interventions are the only option to prevent adverse outcomes in sarcopenic patients [14].

The reduction in the cross-sectional area [15] and nerve fibers (especially the fast-twitch) leads to decreased muscle strength, especially in the lower limbs, linked to longer sitting or lying time [5]. This decrease is associated with a lower capacity to perform daily tasks such as: getting up from a chair, picking up an object from the floor, walking, and climbing stairs [15]. The literature is unanimous, considering that disuse, physical inactivity, and sedentary behaviors constitute determinant causes of muscle strength decrease and muscle loss [16,17]. In the elderly, sarcopenia can reduce the time needed by the muscle to reverse deviation, to maintain mobility, and constitutes a negative factor of quality of life, and may lead to falls and fall-related injuries, requiring costly hospitalization and extended rehabilitation [18].

The European workgroup [19] has established three levels of sarcopenia: (i) the pre-sarcopenia when muscle mass diminishes without strength reduction; (ii) the sarcopenia stage when strength reduction occurs; and (iii) the severe sarcopenia when mass, strength, and performance are diminished. Its multifactorial causes may include disuse, endocrine malfunction, chronic diseases, inflammation, insulin resistance, and nutritional deficiencies [14]. Thus, the multifactorial causes of sarcopenia can be grouped into four domains: metabolic, cellular, vascular, and inflammatory. From those combinations, patients suffering from sarcopenia showed increased intramuscular fat compared to healthy controls [20]. Theoretically, an increased fatty infiltration or transformation of thigh muscles would induce muscle strength to decrease [21]. Whereas evidence of association is scarce, a recent study demonstrated that intermuscular fatty, not the cross-sectional area, can predict muscle strength in thigh muscles [22]. Furthermore, these authors suggest that the interaction between muscular fat and strength could become the basis for a biomarker for muscle quality and function (Figure 1). As skeletal muscle mass, composition, and function are dependent on innervation, and systemic factors [18], examining the decline of neuromuscular function is essential to our understanding of the mechanisms underlying sarcopenia [23].

The pathological cellular mechanisms are based on the reduction of muscle protein syntheses. Accordingly, the power to perform fast movements lessens about 4% per year after 65 years of age. With aging, [24] stated a loss of up to 75% of type II fibers, diminishing functional mobility and muscle power output. Thus, an adequate exercise prescription must be considered for loss of connective tissue elasticity, abilities, and balance.

As aforementioned, sarcopenia is one of the main contributors to functional decline in older people, essentially due to the loss of voluntary strength contraction [25]. Accompanied by muscle loss with age, the number of α-motor neurons synaptic inputs and cortical neurons composing the corticospinal tract also decline [26]; directly impacting muscle fibers excitatory neural inputs. This has two significant implications. First, the decline occurs due to denervation followed by reinnervation. This process results in fewer and larger motor units and slower [27], which is crucial to notice since it contradicts the well-known relationship between fatigability and motor unit size [28]. Second, more frequent activation of larger motor units instead of smaller motor units, and a reduction in the contractile properties, lead to more significant changes in produced muscle force [29] with apparent consequences in balance and gait. Consequently, the muscles responsible for postural responses show larger coactivations in the elderly than in younger persons [30].

The central nervous system also plays a determinant role in the elderly. The vestibular system and the cerebellum are the primary posture and balance control mediators. In contrast, gait control and command are provided mainly by the brainstem, cerebellum, and forebrain [31]. These brain areas are responsible for processing somatosensory information required for movement. Therefore, the importance of cerebral control for balance and gait is evident, with several risk factors affecting them.

Regarding movement, on the one hand, the degeneration of neuronal cells [32] and on the other hand, the adaptation of these cells to minimize the consequences [33] are responsible for a slower initiation [34] and execution movement [35]. Some of these differences in movement time can be caused by how the sensory information is generated and processed [36]. In other words, visual feedback represents a vital role in older populations [37], and compromising this sensory feedback can lead to more movement disturbances.

### 1.2. Correlates of Sarcopenia: Falls and Fall-Related Injuries

Existing literature has shown that sarcopenia is associated with multiple adverse health outcomes, including falls, hospitalization, functional decline, poor quality of life, and in some instances, mortality [38]. As the elderly tend to lose muscle mass and muscle strength, the risk of falling increases. Several studies have reported that one-third of the elderly over 65 years of age fall each year, and the ratio of injuries caused by falls increases with age [11,25]. Fall-related injuries negatively impact up to 40% of the elderly over 75 years old [26] and 50% of those over 80 [27]. These numbers should be worrying and considered a public health problem that the community must carefully consider.

Both imbalance [39] and gait variability [40] increase the probabilities of falling. While both are major risk factors [41] and are considered separate clinical entities, they are often intertwined and depend on several factors, making balance and gait disorders heterogeneous and multifactorial [42]. Thus, assuming that muscle mass and muscle strength are related to gait and balance, sarcopenia constitutes one of the main risk factors for falls among older people.

One of the most severe physical consequences of falls is the fracture of the femur proximal end. About 15% of falls result in dislocations, bruises, and muscle trauma [31], and more than 10% of falls result in bone fractures [32]. Fractures associated with falls in the elderly are a significant morbidity and mortality source as described by several authors [27,29,33]. There is a high probability that the person will not recover functionality to pre-fracture levels [28], frequently, shifting from a situation of independence to dependence on others. This ultimately leads to difficulties in self-care, declining functionality, loss of quality of life, increased comorbidities associated with immobility, fear of falling again, and decreased average life expectancy. As mortality and morbidity are high, these fractures directly impact public health. They are one of the main reasons for dependence, disability, and difficulties for informal caregivers, increasing direct and indirect costs with health.

Besides the physiological consequences, falls have psychological implications, such as losing confidence, restricted activities, and reduced physical functions and social interactions [26,29]. Several authors agree that falls in the elderly have increased in the last 10 years [25,34,35]. Implementing efficient strategies can reduce fall risk, fall-related injuries and reduce the costs associated with health care. In this regard, the relationship between muscle mass and strength and falls is very intimate. The elderly who cannot remain autonomous are more susceptible to falling, increasing the risk of being institutionalized and dependent on others for daily living activities.

### 1.3. Resistance Training, Sarcopenia, and Falls

One of the most used methods for preventing sarcopenia, and consequently falls in the elderly, is physical exercise, following American College of Sports Medicine recommended guidelines [43]. Muscle mass and muscle strength are the main components of physical fitness. Only with acceptable levels of strength can the elderly perform different activities of daily living and have less probability of falling. Climbing stairs, shopping autonomously [36], and engaging in structured physical activity or even sports [37], are daily elements that every elderly should be able to do. Thus, it is vital to examine possible exercise types suitable for older people, especially muscle mass and strength. Resistance training programs in the elderly have demonstrated several benefits in physical fitness components, mainly in muscle mass and strength [40]. This review will provide an overview of the current and relevant literature, evaluate existing resistance training types, and provide evidence-based recommendations for resistance training for the elderly.

Resistance training is a form of periodic exercise whereby internal load or external weights provide progressive stimuli to the skeletal muscles to promote muscle mass and strength [44]. Thus, resistance training includes moving limbs against resistance provided by own body weight, gravity, bands, weighted bars, or dumbbells. The health benefits of resistance training are well-known, and it is recommended for most populations, including adolescents, healthy adults, the elderly, and clinical populations [45]. Resistance training has shown several benefits in older people, such as increasing or maintaining muscle mass and strength [46]. In addition, since muscle mass and strength are correlated with sarcopenia and falls, its maintenance is of utmost importance. Researchers have identified resistance training as one of the most efficient forms of exercise that are most beneficial to older individuals with sarcopenia [47]. The development of optimal exercise programs requires consideration of resistance training.

## 2. Resistance Training: Intensity and Volume

Resistance training is a form of exercise that can increase or maintain muscle mass and muscle strength, which helps older adults preserve their independence and quality of life. It can overcome the loss of muscle mass and strength, build resilience, ease the management of chronic conditions, and reduce physical vulnerability [46]. Resistance training can be done in several ways depending on the physiological and functional or performance goals. The different trainable characteristics of the neuromuscular system include strength, endurance, power, muscle hypertrophy, and motor performance [48]. A list of resistance training types for practical resistance training in older adults is presented below, pointing out intensity and volume.

### 2.1. Strength Training Using Resistance Machines

Traditional strength training is characterized by physical conditioning in which muscles are exercised by being worked against an opposing force (e.g., external load, gravity, elastic band) to increase strength. Executed with slow-speed, strength training is based on muscle contraction against external load, typically resistance machines (e.g., leg press, leg extension). Several systematic reviews and meta-analyses have been conducted on this research topic.

Borde et al. (2015), in their meta-analytic review, found that the most effective dose-response of resistance training on increasing muscle strength and morphology was a frequency of three sessions per week, a training volume of 2–3 sets per exercise, 7–9 repetitions per set, a training intensity ranging from 50 to 70% of the 1RM [49]. Focused on the relationship between strength training and quality of life in the elderly, a meta-analysis conducted by Hart and Buck (2019) found that this type of resistance training had the largest effect on mental health (Effect size (ES) = 0.64, *p* = 0.001) [50]. Peterson and colleagues (2010) observed significant main effects for lower-body strength (i.e., leg press = 31.63 kg (29%); knee extension = 12.08 kg (33%)) and upper-body strength (i.e., chest press = 9.83 kg (24%); lat pull = 10.63 kg (25%)) following strength training interventions [51]. Recently and specifically in elderly in stages of frailty and sarcopenia, a systematic review and meta-analysis of randomized controlled trials conducted by Talar et al. (2021) showed positive and significant changes in handgrip (ES = 0.51, *p* = 0.001) and lower-limb strength (ES = 0.93, *p* < 0.001), and muscle mass (ES = 0.29, *p* = 0.002) [52]. Mañas et al. (2021), in their systematic review and meta-analysis (21 studies) of unsupervised home-based resistance training for community-dwelling older adults RCT, found that significant improvements in lower-limb muscle strength (Hedges’ g = 0.33, *p* < 0.001). The mean frequency of the studies considered was three times per week, with exercises focused mainly on developing muscle resistance and muscle strength and balance [53].

By the position statement from the National Strength and Conditioning Association on resistance training for older adults [46], there is evidence that a strength training performance working towards a frequency of 2–3 times per week, 2–3 sets of 1–2 exercises per major muscle group achieving intensities of 70–85% of 1RM seems to be optimal. Previous meta-analytic reviews [51,52] support this position statement. The most effective frequency for improving muscle strength, increasing the overall quality of life, and decreasing the risk of fall and fall-related injuries is 2–3 times per week. Training should have short sets of moderate-intensity (60–80% of 1RM) exercises focused on major groups; mainly lower limbs since they are significantly related to gait and risk of falling.

There is clear evidence that strength training with resistance machines seems optimal for inducing muscle development or maintenance and reducing fall rates, fear of falling, and increasing quality of life and independence in the elderly. Previously cited research clearly states the importance of prescribing strength training as it is significantly associated with other physical fitness components such as balance, agility, body composition, and flexibility. However, assuming that community-dwelling elderly have access to resistance machines is ambiguous. Most of the cited studies were conducted in laboratory settings or controlled environments, inviting the elderly to participate in control-related studies. Thus, other forms of strength training should be discussed and explored, providing insights for this group to engage in physical exercise and increase their quality of life.

### 2.2. Strength Training Using Bodyweight and Low-Cost Materials

Physical exercise performed in gymnasiums or communities equipped with machines or weights are always an asset. However, this is not a reality in most cases. Exercise programs usually use materials such as dumbbells (e.g., in some instances using water bottles filled with sand), kettlebells, sticks, hoops, own body weight, or elastic bands. Fortunately, numerous practical and efficient exercises can be done under these conditions (i.e., lunges, squats, pull-downs, bicep curls, etc.), when monitored adequately by exercise physiologists.

Skelton et al. conducted one of the first studies using bags of rice and elastic bands as external weight [54]. Exercises resembled everyday tasks such as getting up from a chair, putting weights on the shelves, getting up from the floor, walking in a hallway, among others. The training program consisted of 3 sessions per week for 12 weeks with a volume of 3 × 4 − 8 repetitions, and significant results were found in both strength and power outputs when applying progressive intensity. In the same line, except for the total duration of the training program (10 weeks), the results were significant for the average power regarding the concentric and eccentric movements [55]. Additionally, participants improved functional fitness tests such as walking time, chair stand, and the 8 feet up and go.

Another study using bodyweight and low-cost equipment as resistance showed that the combined exercise and supplementation group had significant results in all three functional fitness tests, namely leg muscle mass, gait speed, and knee extension strength [56]. Since the results were promising, Kim and colleagues conducted new research in which the amino acid supplementation was replaced by tea catechin. Nevertheless, this study evaluated only the functional fitness tests involving the leg muscle mass (%) and gait speed (%). The combined groups (exercise and supplementation) were significant on both fitness tests, and the exercise group only had improvements in gait speed. The control and supplementation groups showed no improvement [57]. Later, the same authors did a similar study but replaced tea catechin with milk fat globule membrane. However, there was a follow-up of 1 month after the intervention in this study. Again, the results showed that the experimental group with combined exercise and supplementation and the isolated exercise group improved physical fitness both in the post-intervention and 1 month post-intervention [58]. All the three previous studies were conducted for three months with two sessions per week. In 2016, researchers did not find significant results from a resistance training protocol. However, this program lasted for ten weeks, and participants only trained twice a week [59]. A year after, the same researchers, Huang, and colleagues (2017) performed a 12-week intervention program with 3-session/week training in sarcopenic obese women. They demonstrated that a simple elastic band resistance training program could reduce fat mass and increase body mineral density, while improvements in muscle mass were scarce [60]. Another study with sarcopenic obese population demonstrated that three months (exercising three times per week) improved muscle mass, physical capacity, and function outcomes (muscle quality of upper and lower extremities, gait speed, timed up and go, timed chair rise, functional forward reach, single-leg stance, and global physical capacity score) compared to the control group [61]. Furthermore, the results were maintained for six months after the three months of resistance exercise intervention. More recently, an extensive nine-month investigation compared resistance training with postural training. Interestingly, the study continued to verify what the literature has been reporting so far with resistance training but demonstrated that postural training isolated is insufficient to help improve muscle mass and strength and static balance in moderate sarcopenic women [62].

## 3. Conclusions

As stated previously, well-designed resistance training programs can prevent falls in older adults. We summarize evidence of benefit for using exercise for fall risk and guide the practical how-to details for effective exercise programs. It is worth mentioning that exercise should not be seen primarily as a treatment but rather as a preventive way to avoid falls, fall-related injuries, and other comorbidities related to advanced sarcopenia.

Exercise physiologists should provide efficient feedback during exercise, focusing elderly to retrain control and activation of the muscles under exertion, posture, and movement patterns, using basic principles and communication styles such as simplification. They should consider a detailed assessment of the recruitment of the muscles during training. Principles of cognitive behavioral therapy, such as goal setting, self-determination, and self-reinforcement, are advised, mainly in elderlies that are new to physical activity. While pain-guided exercises are frequent, the elderly must perform pain-free movements [63].

Professionals should be aware that there is a risk that the elderly may fall while exercising. Prescribed exercises need to be appropriate for each elderly’s physical and cognitive abilities. Furthermore, exercise physiologists should provide advice about the safe conduct of the movement, such as undertaking bodyweight-based or free weight exercises that ought to be performed near firm support (e.g., a wall or table), or preferably seated. Safe storage and application of weights or resistance bands are also necessary. Individual customization of the exercise’s level of difficulty and intensity can ensure the training program is challenging enough to be effective, yet safe [64].

This narrative review provides evidence to support recommendations for practical resistance training in the elderly related to type, intensity, and volume. A properly designed resistance training program with adequate instructions and technique is safe for the elderly. It should include an individualized approach based on existing equipment (i.e., body weight, resistance machines). Existing literature shows that exercise performance towards 2–3 sets of 1–2 exercises per major muscle group and 5–8 repetitions, 2–3 times per week should be recommended, followed by training principles such as periodization and progression. Accordingly, resistance training programs should include fast, and slow-executed exercises focused on developing muscle strength and power since both are related to gait, falls, balance, and quality of life.

Nonetheless, the most efficient resistance training program must be identified, considering the optimal combination between the intensity, volume, and frequency of weekly sessions that can promote muscle adaptations and, in turn, improve the functional capacity in the elderly, thus reducing falls and the risk of falls. It is essential to point out that professionals should adapt resistance training programs according to the elderly’s needs, physiological capacities, equipment availability (e.g., dumbbells), and healthcare investments (e.g., supervised vs. non-supervised exercise programs). Nonetheless, it is generally accepted that exercise programs must be simple, easy to apply for groups, and low cost for higher engagement. Additionally, exercise programs should be fun and display pleasurable feelings during the activity.

## Figures and Tables

**Figure 1 ijerph-19-00874-f001:**
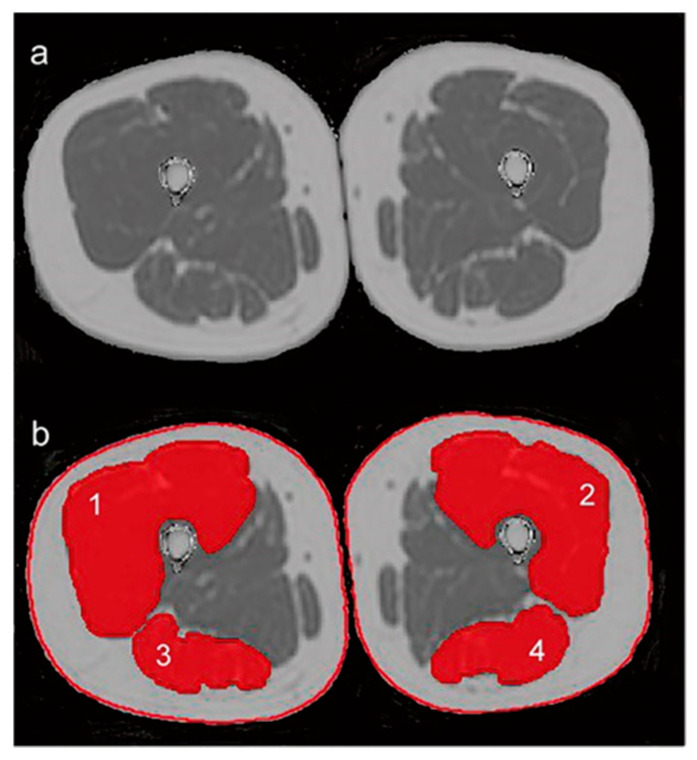
Chemical shift encoding-based water-fat magnetic resonance imaging (MRI) and placement of regions of interest (ROIs). (**a**) Representative proton density fat fraction (PDFF) map. (**b**) PDFF map with superimposition of manually segmented muscle compartments defined as ROIs: (1) right quadriceps muscle, (2) left quadriceps muscle, (3) right ischiocrural muscles, and (4) left ischiocrural muscles. The red lines around the thigh represent the segmentation of the entire thigh contour. Reused from [22] under Creative Commons license.

## Data Availability

Not applicable.

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
