# Peer review of "A Review on Aging, Sarcopenia, Falls, and Resistance Training in Community-Dwelling Older Adults"

_ijerph, 2022, doi:10.3390/ijerph19020874_

Round 1
Reviewer 1 Report
Comments regarding manuscript “A review on resistance training methodologies to reduce Fall-related incidents in the elderly”
The authors presented a literature review aiming to describe the principal therapeutic interventions that help older adults to improve their physical capacities and prevent falls. The authors focused their review on 3 interventions: power and strength training, bodyweight-based exercises, and Whole-body vibration. Additionally, the authors discussed about the relevance of sarcopenia on physical activity levels and health in older adults, about the age-related changes in the Neuromuscular system, and about the relation between muscle strength and fall risk in elderly.
Mayor comments:
- The authors do not define the term resistance training properly, and include several therapeutic interventions that don’t feet exactly with the concept of resistance training, such as Whole-body vibration.
- The evidence has shown diverse results regarding the efficacy of whole-body vibration on muscle strength and physical activity in older adults (efficacy of this intervention in other population is not clear either). It is not clear why whole-body vibration is included in this review. I´m not sure if Whole-bode vibration could be considered a resistance training. Moreover, there is not evidence that could make us think that this intervention contribute to reduce fall incidence.
- The results reported in the review should consider a connection between muscle strength and functional outcomes. In the first part of the section ”power and strength (2.1)” and in the section “Bodyweight-based exercise (2.2” the authors basically reported that the power and strength training improve strength in older adults, without connections with functional outcomes (such as fall incidence… or others).
Minor comments:
- It is clear that sarcopenia could be considered a problem in older adults, and that valid therapeutic interventions are needed to try to reduce the effect of sarcopenia on functional outcomes in elderly. However, in this review the authors do not address this issue int their review.
- The manuscript needs a deep grammar revision. Several grammar and typos mistakes can be found across the manuscript.
- Please add reference in line 80.
Author Response
The authors presented a literature review aiming to describe the principal therapeutic interventions that help older adults to improve their physical capacities and prevent falls. The authors focused their review on 3 interventions: power and strength training, bodyweight-based exercises, and Whole-body vibration. Additionally, the authors discussed about the relevance of sarcopenia on physical activity levels and health in older adults, about the age-related changes in the Neuromuscular system, and about the relation between muscle strength and fall risk in elderly.
R: We appreciate your comments. We have addressed them in full and provide response to each comment point-by-point.
Major comments:
- The authors do not define the term resistance training properly and include several therapeutic interventions that don’t feet exactly with the concept of resistance training, such as Whole-body vibration.
R1.1: Resistance training was defined (see comment R1.1.). Additionally, we eliminated the section 2.3 on whole-body vibration and explored in detail other forms of resistance training.
- The evidence has shown diverse results regarding the efficacy of whole-body vibration on muscle strength and physical activity in older adults (efficacy of this intervention in other population is not clear either). It is not clear why whole-body vibration is included in this review. I´m not sure if Whole-bode vibration could be considered a resistance training. Moreover, there is no evidence that could make us think that this intervention contribute to reduce fall incidence.
R1.2: We agree with your comment as further search on whole-body vibration support your claims. We have eliminated the section 2.3 on whole-body vibration and explored in detail other forms of resistance training.
- The results reported in the review should consider a connection between muscle strength and functional outcomes. In the first part of the section ”power and strength (2.1)” and in the section “Bodyweight-based exercise (2.2” the authors basically reported that the power and strength training improve strength in older adults, without connections with functional outcomes (such as fall incidence… or others).
R1.3: Substantial revisions were made to the entire section related to resistance training.
Minor comments:
- It is clear that sarcopenia could be considered a problem in older adults, and that valid therapeutic interventions are needed to try to reduce the effect of sarcopenia on functional outcomes in elderly. However, in this review the authors do not address this issue int their review.
R1.4: Substantial revisions were made to the entire section related to resistance training.
- The manuscript needs a deep grammar revision. Several grammar and typos mistakes can be found across the manuscript.
R: The entire manuscript was considerably revised to improve the quality of the writing.
- Please add reference in line 80.
R1.6: Reference was added.
Reviewer 2 Report
The authors discussed the published data on resistance training methodologies to reduce fall-related incidents in the elderly. The review paper is considered particularly timely given the growing ageing population worldwide and its social and economic effects. The review could provide a useful guide for medical/and health care practitioners who study the mitigations of falling risks for the ageing population. The review article is well written. Some minor changes are suggested to further improve the manuscript:
- It would be good to summarise the key findings in the abstract
- In the introduction, please list the aim and objectives, in addition, ‘aimed’ and ‘aims’ should be in the same tense.
- What is the sub-heading 1.1?
- In headings, the first letters are sometimes capitalised and sometimes not, please make it consistent.
- Are there any review criteria or strategies that apply? E.g., database, key works, inclusion /exclusion criteria. These should be given before Section2.
Author Response
The authors discussed the published data on resistance training methodologies to reduce fall-related incidents in the elderly. The review paper is considered particularly timely given the growing ageing population worldwide and its social and economic effects. The review could provide a useful guide for medical/and health care practitioners who study the mitigations of falling risks for the ageing population.
R2: We appreciate your comments. We have addressed them in full and provide response to each comment point-by-point.
The review article is well written. Some minor changes are suggested to further improve the manuscript:
- It would be good to summarise the key findings in the abstract
R2.1: Substantial revisions were made in the abstract section. Key findings are provided.
- In the introduction, please list the aim and objectives, in addition, ‘aimed’ and ‘aims’ should be in the same tense.
R2.2: The entire manuscript was considerably revised to improve the quality of the writing.
- What is the sub-heading 1.1?
R2.3: It was a typo. We revised all subheadings accordingly.
- In headings, the first letters are sometimes capitalised and sometimes not, please make it consistent.
R2.4: The entire manuscript was considerably revised to improve the quality of the writing.
- Are there any review criteria or strategies that apply? E.g., database, key works, inclusion /exclusion criteria. These should be given before Section 2.
R2.5: We appreciate your comments. A narrative review is a review of what is considered relevant for the topic and the aim of the review, but without a specified methodological plan as for a systematic review. Thus, it does not follow specific statistical criteria.
Reviewer 3 Report
The purpose of this narrative review is to compile and present information on effective resistance training methods to reduce fall risk in the elderly. The reviewers start the paper with a long discussion about sarcopenia, which seems to dominate the paper. This is followed by a rather surface level discussion and presentation about related resistance training methods.
Strenghts:
This is an important topic and has the potential to be of interest to a wide audience.
Areas of improvement:
1) Translation to English language: There are many areas in the manuscript that would benefit from proof reading. This may be a translation issue, but nonetheless, there are many. Here are some examples from the abstract alone: A) Line 10: the sentence "As aging continues to grow in our society..." is awkward. Maybe, "As the aging population continues to grow..." B) Line 19: Change "state-of-art" to "state of the field" or something similar. C) Lines 25 and 26 are an awkward sentence.
2) The paper is heavy on the discussion about sarcopenia and that is not meant to be a prominent part of the review based on the title. Please condense that section and focus more on expanding the sections on resistance training.
3) The discussion about resistance training methods is lacking. There are a few hand picked examples but not enough substance. The paper would also be improved by a table that displays the most significant references and their major finds along with the condensed methods of resistance training used. This would allow a quick look at factors contributing to intensity and modality, and results in one table. A figure illustrating the benefits of RT in this population would also add a nice summary of major findings.
4) The conclusions have no recommendations based on the literature as to the type of program that is recommended based on the literature.
Author Response
The purpose of this narrative review is to compile and present information on effective resistance training methods to reduce fall risk in the elderly. The reviewers start the paper with a long discussion about sarcopenia, which seems to dominate the paper. This is followed by a rather surface level discussion and presentation about related resistance training methods. Strengths:
This is an important topic and has the potential to be of interest to a wide audience.
R3: We appreciate your comments. We have addressed them in full and provide response to each comment point-by-point.
Areas of improvement:
1) Translation to English language: There are many areas in the manuscript that would benefit from proof reading. This may be a translation issue, but nonetheless, there are many. Here are some examples from the abstract alone: A) Line 10: the sentence "As aging continues to grow in our society..." is awkward. Maybe, "As the aging population continues to grow..." B) Line 19: Change "state-of-art" to "state of the field" or something similar. C) Lines 25 and 26 are an awkward sentence.
R3.1: The entire manuscript was considerably revised to improve the quality of the writing.
2) The paper is heavy on the discussion about sarcopenia and that is not meant to be a prominent part of the review based on the title. Please condense that section and focus more on expanding the sections on resistance training.
R3.2: Substantial revisions were made to the entire section related to resistance training.
3) The discussion about resistance training methods is lacking. There are a few handpicked examples but not enough substance. The paper would also be improved by a table that displays the most significant references and their major finds along with the condensed methods of resistance training used. This would allow a quick look at factors contributing to intensity and modality, and results in one table. A figure illustrating the benefits of RT in this population would also add a nice summary of major findings.
R3.3: Substantial revisions were made to the entire section related to resistance training.
4) The conclusions have no recommendations based on the literature as to the type of program that is recommended based on the literature.
R3.4: Substantial revisions were made to the entire section related to resistance training.
Round 2
Reviewer 3 Report
The manuscript has been greatly improved.
Minor corrections:
1) in the abstract, line 34, and in the conclusion, line 361, there is confusing text regarding set and repetition recommendations.
"...towards 2–3 sets of 1–2 exercises per major muscle group and 3x5-8 repetitions,..."
This is confusing because 2 to 3 sets is stated and then 3x 5-8 repetitions (implying 3 sets of 5 to 8 repetitions). Please edit to keep the 2-3 set range and remove "3x", OR remove the 2-3 sets and keep 3 sets of 5 to 8 repetitions.
2) Lines 366 and 367 are confusing. Is this part of the conclusions?
Author Response
REVIEWER 3
The manuscript has been greatly improved.
R: We appreciate your comments. We have addressed them in full and provide response to each comment point-by-point.
Minor corrections:
1) in the abstract, line 34, and in the conclusion, line 361, there is confusing text regarding set and repetition recommendations.
"...towards 2–3 sets of 1–2 exercises per major muscle group and 3x5-8 repetitions..."
This is confusing because 2 to 3 sets is stated and then 3x 5-8 repetitions (implying 3 sets of 5 to 8 repetitions). Please edit to keep the 2-3 set range and remove "3x" OR remove the 2-3 sets and keep 3 sets of 5 to 8 repetitions.
R3: We appreciate your comments. Edits were made following existing literature (Fragala et al., 2019) and the 2-3 set range was kept.
2) Lines 366 and 367 are confusing. Is this part of the conclusions?
R3: We appreciate your comments. Sentences was revised to clarify the main idea. We believe it is important to acknowledge that “resistance training programs should be adapted according to the elderly’s needs, physiological capacities, equipment availability (e.g., dumbbells), and healthcare investments (e.g., supervised vs. non-supervised exercise programs)” For exercise physiologists to prescribe exercise safe and efficient.